# Sample Efficient Preference Alignment in LLMs via Active Exploration

**Viraj Mehta**[*,1], **Syrine Belakaria**[*,2], **Vikramjeet Das**[3], **Ojash Neopane**[3], **Yijia Dai**[4],
**Ilija Bogunovic**[5], **Barbara Engelhardt**[2], **Stefano Ermon**[2], **Jeff Schneider**[3], **Willie Neiswanger**[6]

[1]TensorZero [2]Stanford, [3]CMU, [4]Cornell, [5]UCL, [6]USC

viraj@tensorzero.com, syrineb@stanford.edu, {vdas,oneopane,schneide}@cs.cmu.edu,
yd73@cornell.edu, i.bogunovic@ucl.ac.uk, {ermon,bengelhardt}@stanford.edu, neiswang@usc.edu

## Abstract

Preference-based feedback is important for many applications in machine learning where evaluation of a reward function is not feasible. Notable recent examples arise in preference alignment for large language models, including in reinforcement learning from human feedback (RLHF) and direct preference optimization (DPO). For many applications of preference alignment, the cost of acquiring human feedback can be substantial. In this work, we take advantage of the fact that one can often choose contexts at which to obtain human feedback to most efficiently identify a good policy, and formalize the setting as an *active contextual dueling bandit* problem. We propose an active exploration algorithm to efficiently select the data and provide theoretical proof that it has a polynomial worst-case regret bound. We extend the setting and methodology for practical use in preference alignment of large language models. We provide two extensions, an online and an offline approach. Our method outperforms the baselines with limited samples of human preferences on several language models and four datasets including two new datasets that we contribute to the literature.

## 1 Introduction

The alignment of foundation models with user preferences has gained unprecedented importance due to the widespread utilization of large language models (LLMs). The established pipeline for alignment in LLMs, as outlined in Stiennon et al. (2020) and Ouyang et al. (2022), comprises two steps given a pretrained LLM. First, in the Supervised Fine-Tuning (SFT) phase, the LLM undergoes fine-tuning via supervised learning with examples demonstrating the desired behavior. In the second step, Reinforcement Learning from Human Feedback (RLHF), a policy generates multiple completions for each conversation prefix (prompt) in a training set; users then give ordinal preferences for the set of completions from a particular prompt. These preferences are used to train a *reward model* via a ranking loss like the Bradley-Terry-Luce model (Bradley & Terry, 1952). Finally, the policy is trained, typically via Proximal Policy Optimization (Schulman et al., 2017), to optimize the reward model while not moving too far from the SFT-trained policy. More recent work (Rafailov et al., 2023), proposed an alternative to RLHF, Direct preference Optimization (DPO), that enables training the LLM policy directly on preference data without using RL and a proxy reward model. As LLMs continue to scale and their areas of application broaden, the number of topics on which we need to align increases, as does the overall scale of human-generated training data requirements. Data annotation for preference-based learning is already incurring a considerable cost for companies that train LLMs. This cost is likely to grow alongside the industry. The issue becomes acute for LLMs in specialized areas such as safety, health, and scientific problems, where the cost of expert feedback can be substantial.

---

[*]Equal contribution

In this work, we take advantage of the fact that we have control over which prompts and completions we provide to human experts to make efficient use of their efforts. Drawing on recent advancements in active exploration for reinforcement learning (Li et al., 2023) and in black-box optimization (Xu et al., 2020), we introduce a method for assessing the value of collecting preferences on specific datapoints, which is both prospective and task-focused. First, we formalize this setting as a *dueling contextual bandit problem* and design an efficient active exploration algorithm that offers polynomial worst-case sample complexity guarantees regarding the policy's performance. Next, we extend these ideas to the alignment setting in LLMs. We show that choosing data for training LLM policies on expert preferences can be targeted by active learning, leading to efficient use of resources under restrictive budgets. In this paper, we build atop the DPO methodology (Rafailov et al., 2023), and develop an acquisition strategy that allows us to actively select preference data based on the DPO training objective. We provide two extensions to our active exploration strategy: the first allows an online learning approach, where data selection and training are based on the model's generations, while the second enables the data selection from offline existing data.

We evaluate our methods on four datasets: the Stanford Human Preferences dataset (Etha-yarajh et al., 2022), the Anthropic Helpful-Harmless dataset (Bai et al., 2022), and two additional datasets which we contribute to the literature: Jeopardy! dataset and Haikus dataset. The Jeopardy! dataset is an extension of an existing dataset from the game show Jeopardy!. It is composed of questions and factual answers to evaluate the ability of an alignment method to avoid hallucinations. The Haikus dataset is composed of instruction prompts to write Haikus with specific details and corresponding examples of satisfactory Haikus. We use three LLMs with different sizes—GPT-2 (Radford et al., 2019), Pythia-2.8B (Biderman et al., 2023), and Llama-3-8B (AI@Meta, 2024)—to showcase a wide range of results and generalization ability. Our full contributions are:

- We formalize the problem of preference data selection as a dueling contextual bandit problem and propose an active exploration algorithm to solve it. We provide a theoretical analysis on the regret bound of our method.

- We propose two extensions of our approach to the LLM preference alignment setting: one using online data and another taking advantage of offline data.

- We contribute two new datasets to the literature: Jeopardy! and Haikus.

- With extensive evaluation, we find that our methods can boost performance by nearly 13% relative to baseline methods when performing preference alignment with a restricted human-feedback sample budget and that they outperform the baselines at avoiding hallucinations on our Jeopardy! dataset.

## 2    Related Work

**Learning from Comparative Feedback**  Reinforcement learning from comparative human feedback has been studied for more than a decade, including work by Fürnkranz et al. (2012), Akour (2014) and, notably, Christiano et al. (2017), which enabled sample-efficient collection of human feedback for deep reinforcement learning (RL) by training a reward model as the RL target. In Appendix D we give a thorough discussion of human feedback in RL and, more recently, LLMs. However, while effective, using human preference feedback comes with substantial costs, which is reflected in state-of-the-art work. For example, Ouyang et al. (2022) emphasize RLHF to improve the alignment of GPT-3 across aspects such as toxicity, hallucinations, and overall quality. Here, the team enlisted the efforts of 40 labelers and worked with a dataset comprising over 100,000 examples labeled by humans.

**Dueling Bandits**   The bandit literature has also explored the effectiveness of comparative feedback—for example, in the "dueling bandit" setting. This was first studied by Yue et al. (2012) in settings where comparative information is relatively easy to extract but absolute rewards (*i.e.*, direct queries) are ill-defined and have no absolute scale. Later, Bengs et al. (2021) surveyed methods that used online learning, where the trade-off with cost of information is most acute, including those used in the online contextual dueling bandit

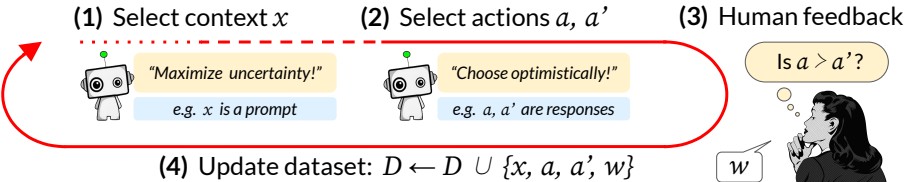

Figure 1: Illustration of the active contextual dueling bandit setting, and its application to sample-efficient preference alignment in large language models.

setting by Dudík et al. (2015). These constraints motivate a kernelized approach that can incorporate the nonlinearities in the models used in practice.

**Active Contextual Bandit Optimization** When there exist distinct phases of learning and then deployment, an agent can often take steps for improved sample efficiency. For example, in a contextual bandit setting, Char et al. (2019) consider the problem where at test time the goal is to perform well on average across a context distribution, while in the learning phase the goal is to choose both contexts and actions for best performance at test-time. The authors propose a multi-task version of Thompson sampling during the training phase. We extend this setting from cardinal to ordinal rewards as is appropriate for comparative feedback.

In Li et al. (2023), the agent queries contexts where the value function is most uncertain and acts optimistically. Combined with least-squares value iteration, this method leads to provable polynomial-sample convergence in the worst-case error of the value function estimate in reinforcement learning in general, and as a corollary the setting from Char et al. (2019) as a special case. This sets the foundation that we will adapt to the comparative feedback setting.

**Uncertainty Estimation in Large Language Models** Estimating the epistemic uncertainty in large language models is still an active area of research and there are few prior works on this topic (focusing specifically on epistemic uncertainty). For example, (Osband et al., 2022) augment existing models with additional layers to model randomness, and subsequently the uncertainty. However performing uncertainty quantification in a parallelized fashion requires a significant memory overhead. To be more amenable to larger models, we instead use a dropout-augmented model to estimate uncertainty (Gal & Ghahramani, 2016), as detailed in Section 5.

In Appendix D, we provide a detailed related work about **alternative contextual bandit methods, and concurrent work on active selection of data in LLMs**.

## 3 Problem Setting

In this paper, we consider a dueling variant of what we denote the Active Contextual Dueling Bandit (ACDB) problem introduced in Char et al. (2019). An instance of this problem is defined by a tuple $(\mathcal{X}, \mathcal{A}, f)$ where $\mathcal{X}$ denotes the context space, $\mathcal{A}$ denotes the action space and $f : \mathcal{X} \times \mathcal{A} \times \mathcal{A} \to [0,1]$ is a preference function so that $f(x, a, a')$ denotes the probability that the action $a$ is preferred to the action $a'$ when the underlying context is $x$. We also define a domain $\mathcal{D} = \mathcal{X} \times \mathcal{A}$. We will design algorithms that operate under the following interaction protocol, which occurs for $T$ time steps. During each time step $t \in [T]$, the agent selects a context $x_t \in \mathcal{X}$ and a pair of actions $a_t, a'_t \in \mathcal{A}$ and observes a binary random variable $w_t \sim \text{Bern}(f(x_t, a_t, a'_t))$ which equals one if $a_t$ is preferred to $a'_t$ and zero otherwise. We assume that the preference function has the form

$$f(x, a, a') = \rho\left(r(x, a) - r(x, a')\right),\tag{1}$$

where $\rho : \mathbb{R} \to [0,1]$ is the *link function* and $r : \mathcal{D} \to \mathbb{R}$ is the *unknown* reward function. Common link functions include the logistic function, which leads to the Bradley-Terry-Luce (BTL) model (Bradley & Terry, 1952) as well as the Gaussian CDF (Thurstone, 1927). We also place additional assumptions on the reward function for our theoretical analysis in the kernelized setting (details in Section 4).

Our goal is to design algorithms that are able to efficiently identify policies with a small suboptimality gap. We define the suboptimality gap of a learner's policy $\pi : \mathcal{X} \to \mathcal{A}$ as

$$\text{SubOpt}(\pi) = \sup_{x \in \mathcal{X}} \left( \sup_{a \in \mathcal{A}} r(x, a) - r(x, \pi(x)) \right). \tag{2}$$

This notion of suboptimality (considered in Char et al. (2019) and Li et al. (2023)) is stronger than notions that look at the expected suboptimality of the final policy when the contexts are sampled from some known distribution. In this work we also use this suboptimality, which looks at the worst-case context for each policy. For the kernelized and LLM settings we provide an explicit instantiation of this problem in Section 4 and Section 5 respectively.

## 4 Active Exploration in the Kernelized Setting

In this section, we describe our first contribution—a theoretically principled algorithm for the ACDB problem—and provide formal guarantees on its performance. To provide the theoretical guarantees, we need to first instantiate our general problem setup by making assumptions on the preference function $f$ (from Eq. (1)). In particular, we specify a class of functions that contain the true unknown reward function. This choice is subtle, as we need to balance the trade-off between our function class's expressiveness and theoretical tractability. Motivated by its theoretical popularity and empirical success, we choose the function class to be a Reproducing Kernel Hilbert Space. While this choice of function class is common in the literature, we make a few additional assumptions to more appropriately accommodate our problem setting.

**The Contextual Borda Function** Before going over our assumptions, we first introduce the *contextual Borda function* $f_r$, which is core to our algorithm. The contextual Borda function generalizes the Borda function introduced in Xu et al. (2020) for dueling-choice optimization, defined as the probability that an action $a$ will be preferred over a random action $a'$ uniformly sampled from the action space. We generalize this definition to the contextual setting as $f_r : \mathcal{D} \to [0, 1]$, where $f_r(x, a) = \mathbb{E}_{a' \sim U(\mathcal{A})} [f(x, a, a')]$ and $U(\mathcal{A})$ is the uniform measure over the action space. It is clear from this definition that $f_r$ and $r$ have the same maximizers.

We now discuss our assumptions. Our first assumption restricts the reward and contextual Borda functions to be 'smooth' in a Reproducing Kernel Hilbert Space (RKHS). Our second assumption relates the reward function to the contextual Borda function.

**Assumption 4.1.** Let $\kappa : \mathcal{D} \times \mathcal{D} \to \mathbb{R}$ denote a positive semi-definite kernel and let $\mathcal{H}_\kappa$ denote its associated RKHS. We assume that $\|r\|_\kappa, \|f_r\|_\kappa \leq B$, where $B$ is a known constant.

Note that this assumption is stronger than the standard assumption, which only requires that $r$ has a bounded RKHS norm. It is difficult to bound the norm of $f_r$ given a bound on the norm of $r$ due to the generality of our setting, which allows for different link functions. We investigate this issue numerically in Appendix C. We find that the norm of the Borda function is almost always smaller than the norm of the reward function for samples drawn from the distribution of basis functions used for experiments in Section 6.1.

**Assumption 4.2.** Let $f_r^*(x) = \max_a f_r(x, a)$ and $r^*(x) = \max_a r(x, a)$. There exists a constant $L_1$ such that for every $x \in \mathcal{X}$, $a \in \mathcal{A}$ we have $\frac{1}{L_1}(r^*(x) - r(x, a)) \leq f_r^*(x) - f_r(x, a)$.

This assumption implies that differences in $r$ will cause a similar magnitude of difference in $f_r$. In fact, when the link function $\rho(\cdot)$ is Lipschitz continuous, it is sufficient for its Lipschitz constant to be at least $1/L_1$ for this condition to hold. We note that this assumption holds for the two most commonly used link functions, the logistic function (Bradley & Terry, 1952) and the Gaussian CDF (Thurstone, 1927).

### 4.1 Methods

At a high level, our approach reduces the dueling feedback problem to contextual optimization over a single action via the *contextual Borda function* introduced above. We then apply

techniques adapted from recent work on active exploration in reinforcement learning to construct a sampling rule and a policy selection rule, which allow us to output a policy with low suboptimality. Broadly, our sampling rule draws contexts which have maximum uncertainty over the Borda 'value function' and then compares the optimistic action with an action sampled uniformly from the action set.

**Estimating the Contextual Borda Function** By design, we can estimate the contextual Borda function using preference data $\{x_t, a_t, a'_t, w_t\}$ by selecting $x_t, a_t$ in an arbitrary fashion and sampling $a'_t$ uniformly at random. For low dimensional settings, our algorithm first estimates the contextual Borda function using standard kernelized ridge regression (KRR) (Rasmussen et al., 2006). We refer the reader to Appendix A for an explicit description of the KRR regression procedure. In Section 5, we explore modifications of our methods for higher-dimensional settings, such as in the case of LLMs. One key feature of KRR is that it provides both an estimate of the contextual Borda function after $t$ observations, $\mu_t(x, a)$, as well as uncertainty quantification of the predictions. Indeed, under Assumptions 4.1 and 4.2 we can show that $|f_r(x, a) - \mu_t(x, a)| \leq \beta \sigma_t(x, a)$ for an appropriately chosen $\beta$ and $\sigma_t(x, a)$ (see Lemma B.1).

**Selecting Contexts and Actions** Our sampling rule builds on top of the one established in Li et al. (2023) and adapts it to our setting. Their algorithm uses uncertainty estimates over the Q-function to guide exploration. Specifically, at each decision step, the algorithm selects the state where the uncertainty in the value of the best action is highest. Formally, the state maximizes the gap between optimistic and pessimistic Q-value estimates. Then, it chooses the action with the highest optimistic Q-value at that state. This approach focuses exploration on parts of the state space where the agent is most uncertain about which action is best, promoting efficient learning of a near-optimal policy. Put simply, the rule is to sample the state with the maximum uncertainty over the value function and then act optimistically. We now present our algorithm, which extends these ideas to the dueling setting via the contextual Borda function $f_r$. For now, we assume that there is a known bonus term $\beta_t^{(r)}$ for all $t$. We can then define upper and lower confidence bounds $\overline{f_r^t}(x, a) = \mu_t(x, a) + \beta_t^{(r)} \sigma_t(x, a)$ and $\underline{f_r^t}(x, a) = \mu_t(x, a) - \beta_t^{(r)} \sigma_t(x, a)$. Our rule is to select a context

$$x_t \in \arg\max_{x \in \mathcal{X}} \left( \max_{a \in \mathcal{A}} \overline{f_r^t}(x, a) - \max_{a \in \mathcal{A}} \underline{f_r^t}(x, a) \right). \tag{3}$$

Here, we are choosing a context that maximizes the difference between the optimistic 'value function' and the pessimistic 'value function' (both of which require a maximization over actions to compute). We then optimistically choose an action

$$a_t \in \arg\max_{a \in \mathcal{A}} \overline{f_r^t}(x_t, a). \tag{4}$$

After repeating this process $T$ times, we output a pessimistic policy against the tightest lower bound we can find, which is the maximizer of all our lower bounds through the optimization process. Put formally, we return $\hat{\pi}_T : \mathcal{X} \to \mathcal{A}$ such that

$$\hat{\pi}_T(x) \in \arg\max_{a \in \mathcal{A}} \max_{t \leq T} \underline{f_r^t}(x, a). \tag{5}$$

We construct the full active exploration algorithm, AE-Borda, given in Algorithm 1.

## 4.2 Theoretical Analysis

In this section, we provide our algorithm's theoretical guarantees. We first introduce the *maximal-information gain*, which plays an important role in our results. The maximum information gain over $t$ rounds, denoted $\Phi_t$, is defined as $\Phi_t = \max_{A \subset \mathcal{X} \times \mathcal{A}: |A| = t} I(r_A + \epsilon_A; r_A)$, where $r_A = [r(x)]_{x \in A}$, $\epsilon_A \sim N(0, \eta^2 I)$, and $I(X; Y) = H(X) - H(X|Y)$ is the mutual information. With this definition, we are now ready to state our result.

**Theorem 4.3.** *Suppose we run Algorithm 1 with* $\beta_t^{(r)} = 2B + \sqrt{2\Phi_t + 1 + \log\left(\frac{2}{\delta}\right)}$, *then, with probability at least* $1 - \delta$, *we have that* $\text{SubOpt}(\hat{\pi}_T) \leq O\left(\frac{L_1}{\sqrt{T}}\left(B + \Phi_T \sqrt{\log \frac{1}{\delta}}\right)\right)$.

---

**Algorithm 1** AE-Borda

---

1: **Input:** kernel function $\kappa(\cdot, \cdot)$, exploration parameters $\beta_t^{(r)}$, number of inital data $n_0$
2: Let $D_{n_0} = \{x_i, a_i, a_i', w_i\}_{i=1}^{n_0}$ for $x_i, a_i, a_i'$ drawn uniformly at random.
3: **for** $t = n_0 + 1, \ldots, T$ **do**
4:     Compute $\mu_t(\cdot, \cdot), \sigma_t(\cdot, \cdot)$ using KRR.
5:     Choose $x_t$ according to Eq. (3).
6:     Choose $a_t$ according to Eq. (4), draw $a_t' \sim U(\mathcal{A})$, and draw $w_t \sim \text{Bern}(f(x_t, a_t, a_t'))$.
7:     Let $D_t = D_{t-1} \cup \{(x_t, a_t, a_t', w_t)\}$.
8: **end for**
9: Output a final policy $\hat{\pi}_T$ according to Eq. (5).

---

We provide the proof and discussion of our theoretical results in the appendix.

## 5 Scaling Active Exploration to Large Language Models

In order to extend our method to the case where $\mathcal{X}$ and $\mathcal{A}$ are both large spaces of sequences as is common in natural language processing, we must address a few **limitations** of the AE-Borda method presented in Section 4.1:

1. The contextual Borda function $f_r$ as defined above is unsuitable for an action space that is extremely large and where most actions are obviously bad (a uniformly sampled sequence of tokens is trivially distinguishable from natural language).

2. Neural network training proceeds in batches, and it would be highly inefficient to label and train on a single example at a time.

3. The uncertainty estimation tools in sequence modeling are more limited than those for explicitly kernelized models, especially due to the memory constraints in training LLMs.

To address these issues, we specialize our method for the LLM setting. In particular, we estimate the uncertainty of our policy using dropout for uncertainty estimation (Gal & Ghahramani, 2016), and perform batched subset selection for our training minibatches. Further, we build atop the foundation presented in Rafailov et al. (2023) on Direct Preference Optimization (DPO), which avoids training a separate reward model; this is primarily due to the fact that we prefer to select datapoints based on the estimated uncertainty of the model used for decision making rather than any proxy.

**Direct Preference Optimization.** Direct Preference Optimization (Rafailov et al., 2023) avoids training a separate reward model based on preferences by instead training the policy directly on pairwise comparison using a loss that optimizes an equivalent objective despite functionally behaving like classification. As with PPO (Schulman et al., 2017), this loss depends on a reference policy, which we take to be the policy derived from the supervised fine-tuning step, $\pi_{\text{SFT}}$. The loss is defined as $\mathcal{L}_{\text{DPO}}(\pi_\theta; \pi_{\text{SFT}}) = -\mathbb{E}_{(x,a,a',w) \sim \mathcal{D}} \big[ \log \rho \big( \gamma(2w - 1) \big( \log \frac{\pi_\theta(a|x)}{\pi_{\text{SFT}}(a|x)} - \log \frac{\pi_\theta(a'|x)}{\pi_{\text{SFT}}(a'|x)} \big) \big) \big]$. Rafailov et al. (2023) shows that optimizing this objective is equivalent to training a PPO policy with a reward function $r(x, a) = \gamma \log \frac{\pi_r(a|x)}{\pi_{\text{SFT}}(a|x)} + \gamma \log Z(x)$ where $\gamma$ is the hyperparameter of PPO scaling the KL penalty, $Z(x)$ is a partition function, and $\pi_r$ is the policy which optimizes the PPO objective.

### 5.1 An Acquisition Function for DPO with Online Selection

Given the first limitation defined above, we propose a generalized contextual Borda function:

$$f_r^\pi(x, a) = \mathbb{E}_{a' \sim \pi(x)} \left[ p(w \mid x, a, a') \right] \tag{6}$$

for a proposal distribution $\pi : \mathcal{X} \to P(\mathcal{A})$. We note that we can recover the original function by setting $\pi(x) = U(\mathcal{A})$. As an extension of the AE-Borda method, we propose the following

selection rule for the proposal distribution:

$$x_t \in \arg\max_{x \in \mathcal{X}} \left( \max_{a \in \mathcal{A}} \overline{f_r^\pi}(x,a) - \max_{a \in \mathcal{A}} \underline{f_r^\pi}(x,a) \right) \tag{7}$$

$$a_t \in \arg\max_{a \in \mathcal{A}} \overline{f_r^\pi}(x_t, a). \tag{8}$$

We can estimate $f_r^\pi$, as well as confidence intervals for any fixed $\pi$, using data where $a'$ is sampled from $\pi$. For LLMs $f_r^{\pi_{\text{SFT}}}$ is a natural choice for $\pi$ since it will provide meaningful samples. Consequently, we can estimate $f_r^{\pi_{\text{SFT}}}$ without offline data by asking a labeler to label $(x, a, a')$ where $a' \sim \pi_{\text{SFT}}(x)$ and then fitting $\pi_\theta$ with the DPO loss. Assuming that we can compute the confidence bounds $\overline{\pi_\theta}(a \mid x)$ and $\underline{\pi_\theta}(a \mid x)$ and using the reward expression in the DPO setting, then we can compute $f_r^{\pi_{\text{SFT}}}$ confidence intervals as follows:

$$\overline{f_r^{\pi_{\text{SFT}}}}(x,a) \approx \frac{1}{n} \sum_{i=0}^{N} \frac{1}{1 + \exp\left( \beta \log \frac{\overline{\pi_\theta}(a_i'|x)}{\pi_{\text{SFT}}(a_i'|x)} - \beta \log \frac{\underline{\pi_\theta}(a|x)}{\pi_{\text{SFT}}(a|x)} \right)} \tag{9}$$

$$\underline{f_r^{\pi_{\text{SFT}}}}(x,a) \approx \frac{1}{n} \sum_{i=0}^{N} \frac{1}{1 + \exp\left( \beta \log \frac{\underline{\pi_\theta}(a_i'|x)}{\pi_{\text{SFT}}(a_i'|x)} - \beta \log \frac{\overline{\pi_\theta}(a|x)}{\pi_{\text{SFT}}(a|x)} \right)}. \tag{10}$$

To estimate the confidence intervals of $\pi_\theta$, we need to estimate its uncertainty. Concretely, we need to address the autoregressive nature of $x$ and $a$. We assume $a$ consists of ordered tokens $t_i$ and $\log \pi(a \mid x) = \sum_{t_i \in a} \log \pi(t_i \mid x, t_1, \ldots, t_{i-1})$. In our method, we use dropout for uncertainty quantification. Specifically, the $m$ dropout masks $d_j$ are integrated into the function $\pi(t_i \mid x, t_1, \ldots, t_{i-1}, d_j)$. During inference, we perform Monte Carlo sampling with dropout, resulting in an ensemble with mean:

$$\mu(t_i \mid x, t_1, \ldots, t_{i-1}) = \frac{1}{m} \sum_{j \in [m]} \log \pi(t_i \mid x, t_1, \ldots, t_{i-1}, d_j). \tag{11}$$

The standard deviation across this ensemble:

$$\sigma(t_i \mid x, t_1, \ldots, t_{i-1}) = \sqrt{\frac{1}{m-1} \sum_{j \in [m]} \left( \log \pi(t_i \mid x, t_1, \ldots, t_{i-1}, d_j) - \mu \right)^2} \tag{12}$$

serves as an approximation for the model's epistemic uncertainty. This technique allows us to capture uncertainty in a computation and memory-efficient manner without compromising model performance. We then define an acquisition function as

$$\alpha(x) = \max_{a \in \mathcal{A}} \overline{f_r^\pi}(x,a) - \max_{a \in \mathcal{A}} \underline{f_r^\pi}(x,a). \tag{13}$$

**An Algorithm for Active DPO.** From here, we use the acquisition function in Eq. (13) to choose points that are maximally informative. We do this in batches in order to respect the constraints of training large models. We address this in a straightforward fashion, fetching a batch of a size much larger than our training batch size, evaluating $\alpha$, and then choosing the top-$b$ elements. We call our full procedure AE-Borda-DPO, and show details in Algorithm 2.

**Extensions to Offline Data.** In some real-world settings, practitioners might have offline datasets $D = \{(x, a, a', w)\}$ with the context and preference actions predefined. In Appendix E we discuss the limitations in using the AE-Borda-DPO algorithm in this offline setting. We then develop and present an *offline active DPO algorithm* for these cases.

## 6 Experiments

We first conduct synthetic experiments to assess the validity of our theory, followed by experiments on LLMs to show the benefits of our method in practice.

---

**Algorithm 2** AE-Borda-DPO

---

1: **Input:** Reference policy $\pi_{\text{SFT}}$, exploration parameter $\beta$, policy constraint weight $\gamma$, batch size $b$, number of iterations $N$
2: **for** $t = n_0 + 1, \ldots, N$ **do**
3:     Draw a large batch of contexts $B = \{x_i\} \sim D$.
4:     **for** $x_i \in B$ **do**
5:         Sample $N$ actions $a'_{ij} \sim \pi_{\text{SFT}}(x_i)$, $m$ actions $a_{ij} \sim \pi_\theta(x_i)$
6:         Estimate $\overline{f_r^{\pi_{\text{SFT}}}}(x, a_{ij})$ and $\underline{f_r^{\pi_{\text{SFT}}}}(x, a_{ij})$ for each $a_{ij}$ using Eq. (9) and Eq. (10).
7:         Compute $\alpha(x_i)$.
8:     **end for**
9:     Let $B_l$ be the top-$b$ elements of $B$ by $\alpha$ value.
10:     **for** $x_i \in B_l$ **do**
11:         Choose action $a_i \in \arg\max_{a_{ij}, j \in [m]} \overline{f_r}(x_i, a_{ij})$
12:         Choose action $a'_{ij}$ uniformly over $j$.
13:     **end for**
14:     Observe preferences labels and add them to $B_l$.
15:     Update the policy $\pi_\theta$ using a gradient step against $\mathcal{L}_{\text{DPO}}$ using $B_l$.
16: **end for**
17: Output $\pi_\theta$

---

### 6.1 Experiments in the Kernelized Setting

In order to assess the validity of our theory we have conducted synthetic experiments that allow us to come as close as possible to the theoretical setting and empirically confirm our results. To do so, we implement regression using the BernoulliGP model provided by GPyTorch (Gardner et al., 2018), using a Matérn kernel.

We test on distributions of synthetic reward functions generated by sampling a random linear combination of Random Fourier Features (Rahimi & Recht, 2007) derived from a squared exponential kernel. For each sampled reward function $r$, we use the Bradley-Terry model with $p(w = 1 \mid a, a', x) = \frac{1}{1+\exp(r(x,a')-r(x,a))}$ to generate comparison data. For each trial we uniformly sample $n_0 = 25$ datapoints and then select data to observe until $T = 500$ total datapoints were collected, via one of three methods: **AE-Borda**: our method, described in Section 4.1, **Uniform-Borda**: uniform sampling of both contexts and actions and **UCB-Borda**: uniform sampling of contexts, along with UCB actions as in AE-Borda.

This last method reduces to the method presented in (Xu et al., 2020) when naively generalized to the contextual setting. All methods have the same test-time behavior of executing the action found by optimizing the pessimistic Borda function estimate for the test context. By optimizing the ground-truth reward function we were able to approximate the optimal policy and therefore estimate the regret of our policy against it. We give an example of the progression of our method for 1D context and 1D actions in Figure 4 as well as a comparison of Uniform-Borda against

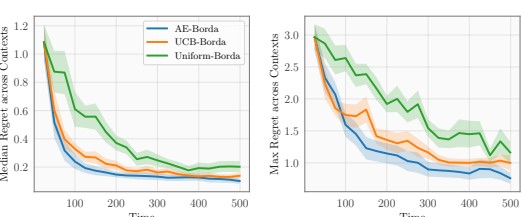

Figure 2: Performance of all methods across 10 random functions $r$ with 1D contexts and 1D actions. The left plot shows the median regret across contexts and the right shows the maximum. Error bands show one standard error.

UCB-Borda in Figure 2. One can see that AE-Borda performs best both on median regret and on the maximum regret, which was the metric of interest in our theoretical analysis.It is clear in Figure 4 that the method is quickly able to concentrate samples in regions that could plausibly be the optimum and that the peaks in the acquisition function over contexts are sensible given the mean and uncertainty estimates of $f_r$. We give a larger set of results showing the progression of AE-Borda in Sec. F.

## 6.2 Experiments using LLMs

To evaluate whether our method is able to improve the selection of data points in DPO, we conduct a set of experiments in which we train LLMs on four datasets using three different models of varying sizes. The goal of our empirical study is to see whether improving the data selection strategy causes the downstream policy to perform better on a given training task. In order to isolate the effect of the data selection method, we varied the selection method while largely keeping our models and training procedure consistent. In all the experiments in this section, we compare two methods: **AE-Borda-DPO**, the method we presented in Section 5.1, **Uniform-DPO**, the method from Rafailov et al. (2023), selecting batches of contexts and their actions uniformly at random, and **OAT-DPO**, the method from Liu et al. (2024). OAT originally uses a separate reward model. For fair comparison, we adapt it to the DPO setting and use the implicit reward model. The code for our approach is publicly available github.com/belakaria/active-llm-alignment.

In our training pipeline, we first train a baseline model using supervised fine-tuning (SFT) on all the supervised data. We add a dropout layer before the penultimate linear layer for our uncertainty estimation mechanism and fine-tune it with the dropout activated. Next, we train each of our model-dataset pairs on a fraction of the prompts while generating the preference samples from the policy and collecting the preference label using an oracle. We use GPT-4o-mini as our training-time oracle.

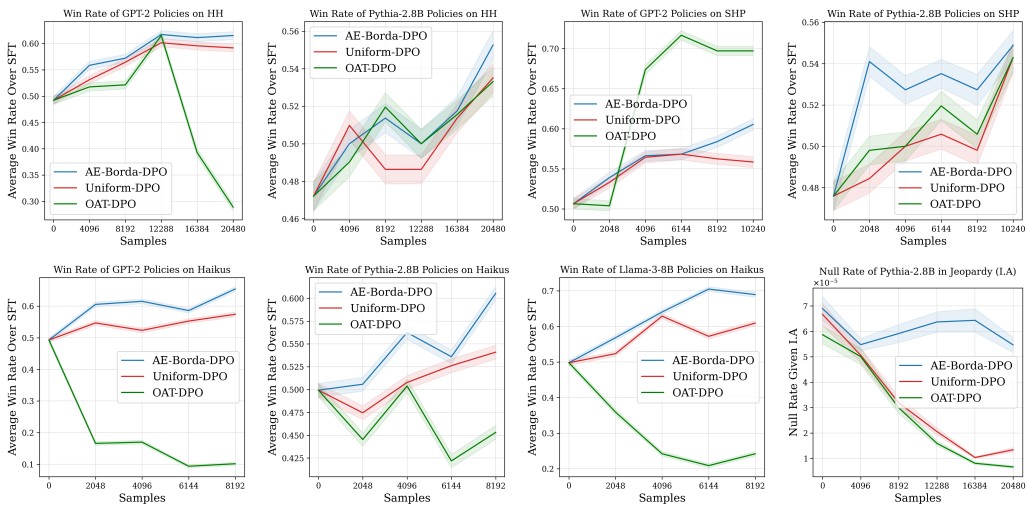

Figure 3: **Active Exploration for DPO in LLMs.** For multiple models and datasets we compare AE-Borda-DPO vs Uniform-DPO (Rafailov et al., 2023) and OAT-DPO (Liu et al., 2024). In the first six plots we show the average win rate over supervised fine-tuning (SFT), and in the final two plots (on the Jeopardy! dataset) show Null Rate given an incorrect answer (I.A.).

We evaluate both methods on four different datasets. The first two, Anthropic Helpful-Harmless (HH) (Bai et al., 2022) and Stanford Human Preferences (SHP) (Ethayarajh et al., 2022), are taken from the literature. HH contains examples of two kinds: situations where an assistant needs to be helpful to a user asking a reasonable question and situations where an assistant should prioritize being harmless as the user is requesting a harmful action. All completions in HH are machine-generated. SHP is a dataset of Reddit posts with comments in 18 different categories and consists of a broader range of human-generated text, but doesn't have the inherent tradeoff of HH. The third dataset is a Haikus dataset that we contribute to the literature. It is composed of 28,000 instruction prompts to write Haikus with specific details and corresponding examples of satisfactory Haikus.

We also introduce a new Jeopardy! dataset that includes a preference structure capturing aspects of the game while also addressing the question of LLM hallucination. We augment a dataset of 217,000 Jeopardy! questions and answers from HuggingFace (Wolf et al., 2023) with a plausible incorrect answer, using GPT-3.5. As in the game show, where points are

deducted for an incorrect response, we embed in the training data that a correct answer is preferred to an abstention (the empty string), and both are preferred to the incorrect answer.

We evaluate performance by checking the rate at which the policy produces answers that are preferred to those generated by the initial SFT policy. To assess the win rate, we use GPT-4o as a judge. We evaluate policies' performance every 2048 training samples for the SHP and Haikus datasets, and every 4096 samples for the Jeopardy! and HH datasets. To mitigate position bias, each pair of responses is evaluated twice, reversing their order in the second evaluation. We then report the average win rate across both evaluations.

We use GPT-2, Pythia-2.8-B, and Llama-3-8B as policies paired with the above datasets to provide a wide range of results with varying model sizes and capabilities. We provide extensive evaluation using 8 different experiments: GPT-2 trained on HH, SHP, Haikus, and Jeopardy! datasets, Pythia-2.8 trained on HH and Jeopardy! datasets and Llama-3-8B trained on SHP and Haikus datasets.

In Figure 3, we see that our method AE-Borda-DPO, especially in the later part of our training run, consistently outperforms the standard DPO baseline that samples uniformly. We also notice that it outperforms the OAT baseline in most experiments. We believe this to be due to our acquisition function $\alpha$, which accounts for the structure of the decision-making problem in choosing which point to query. While we do find our results to be somewhat noisy with high standard error, due to the computational expense of these trials and also the training oracle cost, we were not able to run each experimental baseline for a very large number of seeds to further reduce uncertainty in our results.

In the appendix, we provide the win rate for the Jeopardy! experiments. In Jeopardy!, we do not aim for our models to learn to provide correct answers at a higher rate through DPO training. This is because trivia is intended not to generalize easily; in other words, it is difficult to imagine learning that the third US president was Jefferson given training examples of the first two. Instead, we evaluate policies for this dataset on the rate at which they abstain for questions ("null rate") where they counterfactually would have been correct vs where they would have been incorrect. Ideally, the policy learned would *always* abstain where it would have been incorrect and *never* abstain where it would have been correct. Naturally, this is an important goal in the alignment of LLMs and we hope to provide a straightforward benchmark for this effort.

For the Jeopardy! dataset, we checked the probability of an empty generation and whether it was the most likely token. We generated a nonempty sequence in order to see whether the generated answer was correct, including as a counterfactual in the cases where the method would have abstained. We plot this in Figure 3 (final two plots), where we see that AE-Borda-DPO learns to abstain from answering questions more often when the model would have given the incorrect answer. However, we observe that the null rate in the online setting is low for both approaches. It is worth noting that the null rates given incorrect answers are higher when using offline data (results shown in Appendix E). This observation indicates that with training on model generations, it is harder to learn to abstain. This is mainly due to the fact that it is not possible to explicitly embed the abstaining preference when we do not have direct control over the preference data. In the appendix, we also provide null rates given the correct answer. We note that the abstaining rate is nearly null for all methods in the case where they would have been correct, which is the desired behavior.

## 7   Summary

In this work, we addressed the problem of how to select contexts and actions at which to obtain human preferences, such that a reinforcement learning agent learns most efficiently. We focus on this problem setting in the context of preference alignment in large language models (LLMs), where collecting data from humans is expensive. The methods developed in this work show promise in reducing these costs. We also make a theoretical contribution by giving guarantees on worst-case regret. While our computational study was constrained by the resources available, given our method's promising results, we hope to scale up our experimental campaign to greater numbers of GPU resources and DPO steps in the future, to see how our methods perform with larger computational budgets.

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

# Appendix

## A  RKHS Regression

At step $t$, we have data $\{(x_1, a_1, a_1', w_1), \ldots, (x_t, a_t, a_t', w_t)\}$. The kernel ridge regression estimate is defined by,

$$\mu_t = \arg\min_{f \in \mathcal{H}} \sum_{i=1}^{t} (f(x_i, a_i) - w_i)^2 + \lambda \|f\|_{\mathcal{H}}^2 \,. \tag{14}$$

Denote by $\boldsymbol{w}_t = [w_1, \ldots, w_t]^T$ the vector of observations, $(K_t)_{i,j=1,\ldots,t} = k(x_i, a_i, x_j, a_j)$ the data kernel matrix, and $k_t(x, a) = [k(x, a, x_1, a_1), \ldots, k(x, a, x_t, a_t)]^T$ the data kernel features. We then have

$$\mu_t(x, a) = k_t(x, a)^T (K_t + \lambda \mathbf{1}_t)^{-1} \boldsymbol{w}_t \,. \tag{15}$$

We further have the posterior variance $\sigma_t(x, a)^2$ that determines the width of the confidence intervals,

$$\sigma_t(x, a)^2 = k(x, a, x, a) - k_t(x, a)^T (K_t + \lambda \mathbf{1}_t)^{-1} k_t(x, a) \,. \tag{16}$$

## B  Proof of Theorem 4.3

In this section we will prove our main Theorem, 4.3.

**Proof Overview.** First, we use standard results on KRR to show that our choice of $\beta^{(r)}$ guarantees that our confidence bands contain $f^r(x, a)$ with high probability simultaneously for all $t$ and $x, a \in \mathcal{X} \times \mathcal{A}$. Next, we use Assumption 4.2 to show that the suboptimality of the pessimistic policy induced by our estimated contextual Borda function is small whenever we are able to estimate the contextual Borda function well. Finally, we conclude the proof by showing that our sampling rule indeed allows us to estimate the contextual Borda function well. The full proof can be found in Appendix 4.3.

**Concrete Performance Bounds.** To more concretely understand the performance of our algorithm, we instantiate our results for two commonly studied kernels: the linear and squared-exponential. For both of these settings, the scaling of the information gain is well known (see for example (Srinivas et al., 2010)). In the linear setting, we have that $\Phi_T = O(d \log T)$ leading to a bound of $O\left(\frac{L_1}{\sqrt{T}} (d \log T)\right)$. For squared exponential kernels we have $\Phi_T = O\left(\log(T)^{d+1}\right)$ leading to a suboptimality bound of $O\left(\frac{L_1}{\sqrt{T}} \left(\log(T)^{d+1}\right)\right)$.

When compared to existing results for dueling bandits (Xu et al., 2020) as well as standard bandits (Chowdhury & Gopalan, 2017), we see that our suboptimality bounds match, thus showing that our algorithm is able to achieve the same performance under a stronger performance metric.

**Proof.** The overall strategy of the proof is to use our Lipschitz assumption on the link function (more precisely, the relative Lipschitzness of the reward $r$ and the Borda function $f_r$) in order to go to the Borda function, which we can directly model from data. Then, we use our selection criteria as well as confidence bounds taken from Chowdhury & Gopalan (2017) and convergence rates taken from Kandasamy et al. (2019) in order to complete the argument. We give these cited results as lemmas in what follows.

In order to attain a particular policy performance with probability $1 - \delta$, we must bound the error of the estimates given by our KRR process for a particular confidence level. In order to do so, we adapt the result from Chowdhury & Gopalan (2017), Theorem 2.

**Lemma B.1.** *Let* $\beta_t^{(r)} = 2||f_r||_\kappa + \sqrt{2(\Phi_{t-1}(\mathcal{X} \times \mathcal{A}) + 1 + \log(2/\delta))}$. *Then with probability* $1 - \delta$ *we have for all time* $t$ *and any point* $(x, a) \in \mathcal{X} \times \mathcal{A}$,

$$|\mu_{t-1}(x, a) - f_r(x, a)| \leq \beta_t^{(r)} \sigma_{t-1}(x, a).$$

*Proof.* To prove this result, we will verify that all the conditions from Theorem 2 of Chowdhury & Gopalan (2017) hold. Recall Assumption 4.1 which states that $||f_r||_\kappa \leq B$. Next, we observe that since $a_t' \sim U(\mathcal{A})$ (independent of everything else), we have that $\mathbb{E}[w_t \mid \mathcal{F}_{t-1}] = f_r(x_t, a_t)$, where $\mathcal{F}_t = \rho\left(\{(x_s, a_s, a_s', w_s)\}_{s=1}^t\right)$ is the filtration generated by the past observations. Additionally, since $w_t \in \{0, 1\}$ and $x_t, a_t$ are both $\mathcal{F}_{t-1}$ measurable, we see that $w_t$ can be written as

$$w_t = f_r(x_t, a_t) + \eta_t,$$

where $\eta_t$ is $\mathcal{F}_{t-1}$-conditionally subGaussian. Therefore, we have met all the necessary conditions, and we can apply Theorem 2 of Chowdhury & Gopalan (2017) which gives us the desired result. □

This lemma jointly bounds the modeling error over the Borda function for all time $t$ though it introduces a dependence on the RKHS norm of $f_r$. This dependence is inherited from prior work, but we empirically study the relationship between the RKHS norm of a particular reward function and that of the associated Borda function in Section C.

We also adapt a result from Lemma 8 of Kandasamy et al. (2019) in order to understand the convergence of our uncertainty function $\sigma_t$.

**Lemma B.2.** *Suppose we have* $n$ *queries* $(q_t)_{t=1}^n$ *taken from* $\mathcal{X} \times \mathcal{A}$. *Then the posterior* $\sigma_t$ *satisfies*

$$\sum_{q_t} \sigma_{t-1}^2(q_t) \leq \frac{2}{\log(1 + \eta^{-2})} \Phi_n(\mathcal{X} \times \mathcal{A}).$$

Lemma B.2 gives us a handle on how quickly we can expect the uncertainty function to shrink as additional datapoints are observed.

Now that we have lemmas B.1 and B.2 in place, we can proceed to the proof of the main result.

*Proof.* In this proof, we condition on the event in Lemma B.1 holding true. Given that occurence, we can say the following for every $x \in \mathcal{X}$.

$$\max_{a \in \mathcal{A}} r(x,a) - r(x, \hat{\pi}_T(s)) \overset{\text{Assumption 4.2}}{\leq} L_1 \left( \max_{a \in \mathcal{A}} f_r(x,a) - f_r(x, \hat{\pi}_T(x)) \right) \tag{17}$$

$$\overset{\text{Lemma B.1}}{\leq} L_1 \left( \max_{a \in \mathcal{A}} f_r(x,a) - \max_{t \in [T]} \underline{f_r^t}(x, \hat{\pi}_T(x)) \right) \tag{18}$$

$$\overset{\text{Def. of } \hat{\pi}_T}{=} L_1 \left( \max_{a \in \mathcal{A}} f_r(x,a) - \max_{a \in \mathcal{A}} \max_{t \in [T]} \underline{f_r^t}(x,a) \right) \tag{19}$$

$$= L_1 \min_{t \in [T]} \left( \max_{a \in \mathcal{A}} f_r(x,a) - \max_{a \in \mathcal{A}} \underline{f_r^t}(x,a) \right) \tag{20}$$

$$\overset{\text{Lemma B.1}}{\leq} L_1 \min_{t \in [T]} \left( \max_{a \in \mathcal{A}} \overline{f_r^t}(x,a) - \max_{a \in \mathcal{A}} \underline{f_r^t}(x,a) \right) \tag{21}$$

$$\overset{\text{Def. of } x^t}{\leq} L_1 \min_{t \in [T]} \left( \max_{a \in \mathcal{A}} \overline{f_r^t}(x^t,a) - \max_{a \in \mathcal{A}} \underline{f_r^t}(x^t,a) \right) \tag{22}$$

$$\overset{\text{Def. of } a^t}{\leq} L_1 \min_{t \in [T]} \left( \overline{f_r^t}(x^t, a^t) - \underline{f_r^t}(x^t, a^t) \right) \tag{23}$$

$$\leq \frac{L_1}{T} \sum_{t=1}^{T} \left( \overline{f_r^t}(x^t, a^t) - \underline{f_r^t}(x^t, a^t) \right) \tag{24}$$

$$= \frac{L_1}{T} \sum_{t=1}^{T} 2\beta_t^{(r)} \sigma_t(x^t, a^t) \tag{25}$$

$$\overset{\beta_t^{(r)} \text{ is increasing}}{\leq} \frac{2L_1 \beta_T^{(r)}}{T} \sqrt{\left( \sum_{t=1}^{T} \sigma_t(x^t, a^t) \right)^2} \tag{26}$$

$$\overset{\text{Cauchy-Schwarz}}{\leq} \frac{2L_1 \beta_T^{(r)}}{T} \sqrt{T \sum_{t=1}^{T} \sigma_t^2(x^t, a^t)} \tag{27}$$

$$\overset{\text{Lemma B.2}}{\leq} \frac{2L_1 \beta_T^{(r)}}{\sqrt{T}} \sqrt{C_1 \Phi_T} \tag{28}$$

$$\overset{\text{def of } \beta_T^{(r)}}{=} \frac{2L_1}{\sqrt{T}} \left( 2B + \sqrt{2(\Phi_{t-1} + 1 + \log(2/\delta))} \right) \sqrt{C_1 \Phi_T} \tag{29}$$

$$= O \left( \frac{L_1}{\sqrt{T}} \left( B + \Phi_T \sqrt{\log \frac{1}{\delta}} \right) \right). \tag{30}$$

$\square$

## C RKHS norms of $r$ and $f_r$

In order to understand the dependence of our estimation bound on the RKHS norm $||f_r||_\kappa$, we ran numerical experiments on sampled reward functions. For a variety of context and action dimensions, we sampled 1000 reward functions as in Section 6.1 and numerically approximated their RKHS norms. We also made a Monte-Carlo estimate of the Borda function $f_r$ for each of the reward functions sampled and numerically approximated its RKHS norm. To do this, we uniformly sample 1,000 points $x_i$ from the input space, compute the regularized kernel matrix $K$ for this set $x_i$, solve the KRR problem $K\alpha = f(x)$ for $\alpha$. Then we compute the quadratic form $\sqrt{\alpha^T K \alpha}$ as an estimate of the RKHS norm.

In Table 1, we present the results of comparing the RKHS norms of 1000 reward functions and their associated Borda functions sampled as in Section 6.1. A 'win' was counted when the Borda function had smaller RKHS norm and a 'loss' otherwise. The win margin is the average difference in RKHS norms of the reward and Borda functions, with a positive

| Context Dimension | Action Dimension | Win Rate | Win Margin |
|---|---|---|---|
| 0 | 1 | 0.16 | -6.3 |
| 1 | 1 | 0.89 | 5.1 |
| 1 | 3 | 1 | 21.4 |
| 3 | 1 | 1 | 21.5 |
| 3 | 3 | 1 | 38.7 |
| 10 | 10 | 1 | 19.6 |

Table 1: Comparison of RKHS norms of reward functions and associated Borda functions

value when the Borda function was of smaller norm. It is clear here that in general (though not always) the RKHS norm of the Borda function $f_r$ for a particular reward function $r$ is smaller than the RKHS norm of the reward function $r$ itself. This relationship seems to grow stronger as the input dimensionality of the reward function grows larger.

## D   Additional Related Work

In this section, we discuss additional related work, including alternative contextual bandit methods, uncertainty estimation in large language models, and concurrent work on active selection of data in LLMs.

**Human feedback in RL and LLMs**   Here we discuss additional related work on human feedback in reinforcement learning, and more recently, in LLMs. As described in Section 2, Christiano et al. (2017) enabled sample-efficient collection of human feedback for deep reinforcement learning (RL) by training a reward model as the RL target. This technique showed significant performance benefits in practice; for example, in the Atari test case (Mnih, 2013), where naive deep RL would have necessitated thousands of hours of gameplay, they accomplished superior performance with just 5,500 or several hours of human queries. More recently, human preference feedback has also been used more recently to improve the performance of LLMs. For example, many recent approaches have demonstrated the effectiveness of using human feedback to enhance LLM stylistic continuation (Ziegler et al., 2019), text summarization (Stiennon et al., 2020), translation (Kreutzer et al., 2018), semantic parsing (Lawrence & Riezler, 2018), review generation (Cho et al., 2018), and evidence extraction (Perez et al., 2019). In particular, the work by (Bai et al., 2022) places focus on improving model reliability and robustness by incorporating human feedback to gauge the helpfulness or harmfulness of its responses. However, while effective, the integration of human feedback comes with substantial costs. For example, Stiennon et al. (2020) achieved substantial enhancements over baseline methods but required the generation of summaries for 123,169 posts from the TL;DR dataset, a task performed by a large team of labelers from crowdsourcing platforms. This heavy resource requirement is reflected in state-of-the-art work. Ouyang et al. (2022) emphasizes RLHF to improve the alignment of the GPT-3 model across aspects such as toxicity, hallucinations, and overall quality. Here, the team enlisted the efforts of 40 labelers and worked with a dataset comprising over 100,000 examples labeled by humans.

**Concurrent work on active learning in LLMs**   Concurrently with our work, there has been recent releases of papers related to active data selection for LLMs, which we cover in this section. Note that these papers are predominantly recent and yet unpublished work, released on preprint servers, some of which build on our method and setting. For example, Das et al. (2024); Ji et al. (2024) builds on our active contextual dueling bandit setting. Das et al. (2024), aiming to develop a method that yields improved theoretical guarantees with reduced assumptions. Zhang et al. (2024) proposed a version of DPO using bilevel optimization to optimistically bias towards potentially high-reward responses, though does not use an explicit uncertainty estimate. Xiong et al. (2024) develops an online exploration algorithm as well as a rejection sampling method for offline settings, framing the problem as a reverse-KL regularized contextual bandit problem. Muldrew et al. (2024) propose an

active learning method for DPO, based on the predictive entropy of LLM predictions as well as uncertainty given by the (implicit) reward model. Xie et al. (2024) presents a method that performs DPO with an exploration bonus for improved efficiency. Finally, Hübotter et al. (2024) work on a method for active selection of examples for fine-tuning of LLMs using active data selection, for a (single) given prompt at test time.

# E   Active DPO Using the Reward Function and Offline Data

In this section, we start by proposing another active learning acquisition function based on the reward model. Then we provide a discussion contrasting the real use cases of active learning using online data generated from the policy and the synthetic setting where we can use existing offline benchmarks to evaluate active learning methods.

We propose a new acquisition function that uses the confidence interval of the reward function instead of the generalized Borda function that operates based on the preference model. Using the reward model provides an intuitive solution in RLHF in general and DPO in particular, since the goal is to learn a policy that generates high-reward answers. We can approximate the confidence interval for $r$ ($\overline{r}$ and $\underline{r}$) using the reward expression as the ratio of the policies as defined in the DPO paper. Given the uncertainty estimation method described in section 5.1, we can compute our upper and lower bounds as

$$\overline{r}(x, a) = \sum_{t_i \in a} \mu(t_i \mid x, t_1, \ldots, t_{i-1}) + \beta\sigma(t_i \mid x, t_1, \ldots, t_{i-1}) - \log \pi_{\text{SFT}}(a \mid x),$$

$$\underline{r}(x, a) = \sum_{t_i \in a} \mu(t_i \mid x, t_1, \ldots, t_{i-1}) - \beta\sigma(t_i \mid x, t_1, \ldots, t_{i-1}) - \log \pi_{\text{SFT}}(a \mid x),$$

for an uncertainty parameter $\beta > 0$. Here, we define an acquisition function as:

$$\alpha(x) = \max_{a \in \mathcal{A}(x)}\overline{r}(x, a) - \max_{a \in \mathcal{A}(x)}\underline{r}(x, a). \tag{31}$$

In this equation, $\alpha(x)$ is the uncertainty of the state-value function according to $x$. In choosing the states where the potential for error in the value achieved is largest, the agent can learn to behave well in those places. This criterion is similar to that in (Li et al., 2023) and provides similar guarantees to ours for max-regret in the active contextual bandit setting. In situations like ours where we are using fixed offline datasets, we set $\mathcal{A}(x)$ in Eq. (31) to the set of available responses for a particular action; otherwise, we use $\mathcal{A}(x) = \mathcal{A}$.

**An algorithm for active RLHF/DPO**   From here, we use the acquisition function in Eq. (31) in order to choose points that are maximally informative. We must do this in batches in order to respect the constraints of training large models. We address this in the naive fashion, pulling a larger batch of some size, evaluating $\alpha$ and then choosing the top-$b$ elements in order to address this criterion. We refer to our full procedure as AE-DPO, and give a description in Algorithm 3. Though the use of this acquisition function means that we lose

---

**Algorithm 3** AE-DPO

1: **Input:** Reference policy $\pi_{\text{SFT}}$, exploration parameter $\beta$, policy constraint weight $\gamma$, batch size $b$, number of iterations $N$
2: **for** $t = n_0 + 1, \ldots, N$ **do**
3:     Draw an unlabeled batch $B_u = \{(x_i, a_i, a_i')\} \sim D$.
4:     Evaluate $\alpha(x_i)$ and let $B_l$ be a batch of the top-$b$ elements of $B_u$ by $\alpha$ value.
5:     Collect labels $r_i$ and add them to $B_l$.
6:     Update the policy $\pi_\theta$ (initialized from the ref.) using a gradient step against $\mathcal{L}_{\text{DPO}}$ using $B_l$.
7: **end for**
8: Output $\pi_\theta$

---

the theoretical guarantees given by the Borda function, we assert that given the rates of convergence associated with kernelized approximations of neural net architectures, we are not giving up strong guarantees in this setting.

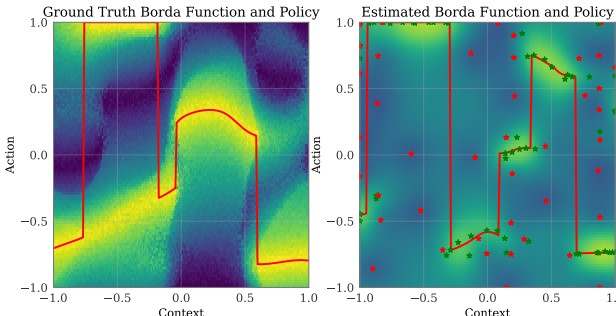

Figure 4: Visualizing the Borda function estimate with 100 data points. Left: the ground truth contextual Borda function $f_r$ (red line is the optimal policy). Right: the mean of our posterior estimate of $f_r$ (red line is the best policy estimate). Red dots are queries where $w_t = 0$ and green are where $w_t = 1$. For a full version, see Fig. F.

**Algorithms Evaluation on Offline and Online Preference Data**   The real use case scenario of active learning for preference data would be on pairs of data points that differ from any data used during the SFT step (otherwise the SFT labels would be by definition always the preferred action). Therefore, the online approach we proposed in the main paper with AE-Borda-DPO reflects the real use case of active learning. However, prior work on preference learning (Rafailov et al., 2023) makes use of existing offline datasets with labeled preferences (e.g. Stanford Human Preferences Dataset (Ethayarajh et al., 2022)). Concurrent active learning work () also uses these datasets to evaluate their approaches in hypothetical scenarios for easier training and evaluations. We do the same and use the offline datasets to evaluate the AE-DPO method. However, evaluating our original AE-Borda-DPO approach on the offline dataset was not feasible due to the acquisition function definition that assumes that the actions selected are not seen by the SFT policy before. Assuming perfect training of the policy on the SFT data (also referred to as chosen answers $a$), if we evaluate our acquisition function on the existing pair of data, we will have:

$$\pi_{\text{SFT}}(x) = \begin{cases} a & \text{if } w = 1 \\ a' & \text{else} \end{cases}$$

for every point in our dataset. And if we were to regress on $f_r^{\pi_{\text{SFT}}}$, all our labels would be 0 (since $\pi_{\text{SFT}}$ is defined as the policy that wins on the dataset). Though this detail is a result of properly defined assumptions, it prevents us from using the offline labeled data to evaluate AE-Borda-DPO. However, this issue does not occur in the definition of the AE-DPO acquisition function, so we use the offline data for its evaluation.

# F   Additional Experiments for Kernelized Setting

In Figure 5, we depict the progress of the AE-Borda method as it continually acquires data. One can see that the estimated optimal policy (red, second row) converges to a function quite similar to the ground truth (red, first row) as more data is collected. In addition, it is clear that the selection criterion targets parts of the domain which are relevant to policy learning while avoiding obviously bad regions. We also see in the fourth row that the uncertainty over the value function decreases relatively smoothly across the context space, supporting the idea that our method controls max-regret effectively.

# G   Additional Experiments for LLMs: AE-Borda-DPO on Jeopardy!

In Figure 6, we provide the full experiments on Jeopardy! dataset and include the win rate. As discussed in the main paper, the win rate does not improve significantly for any of the methods but the null rate given incorrect answers is higher for AE-Borda-DPO.

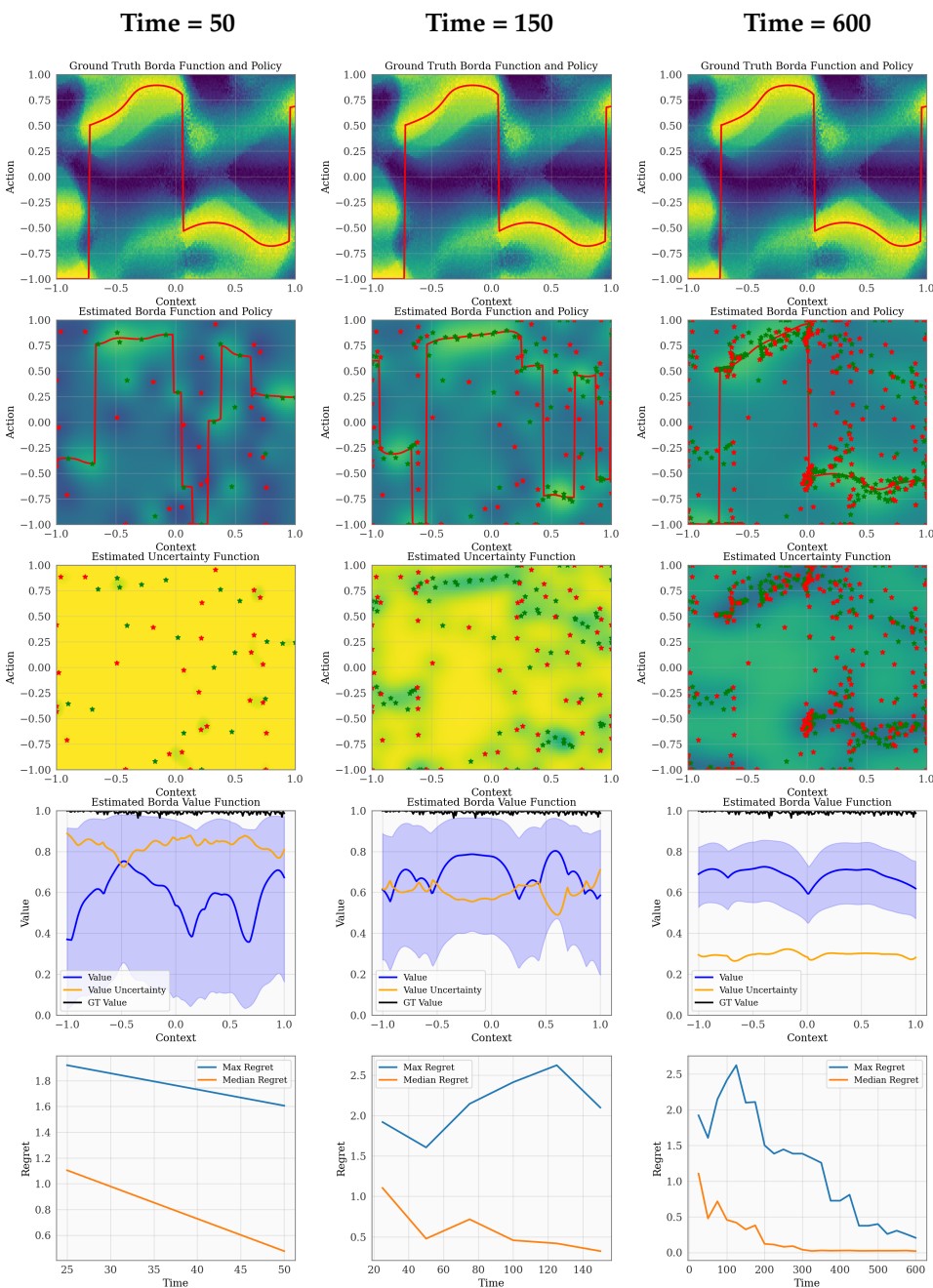

Figure 5: Progress of AE-Borda across 50, 150, and 600 datapoints. From the top downwards, the charts show the ground truth function, the mean of the posterior estimate of $f_r$, the uncertainty function, the estimate of the value function as well as the acquisition function given in Eq. (3), and the regret over time.

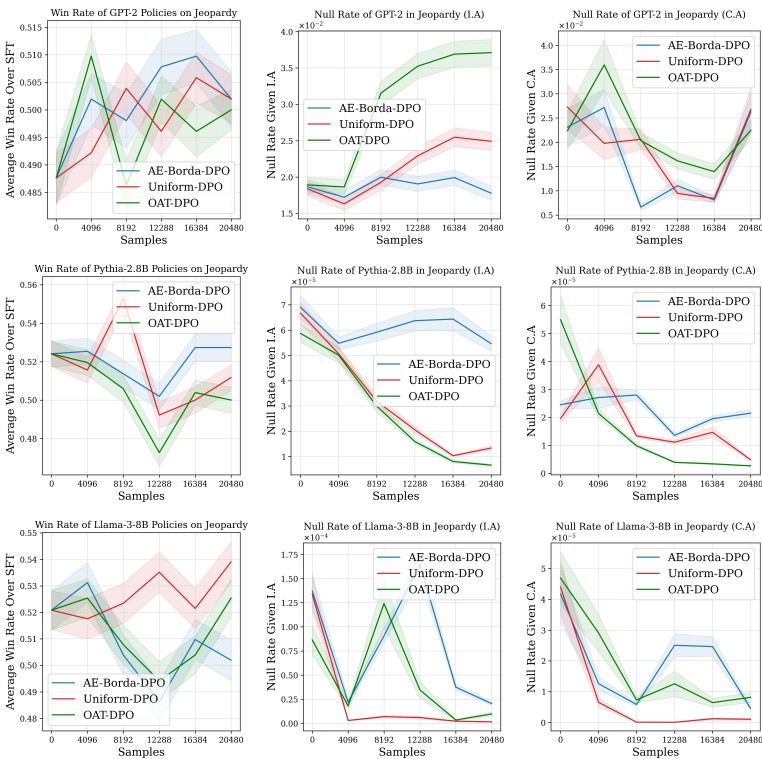

Figure 6: **Active Exploration for DPO in LLMs.** Experiments run on Jeopardy dataset. The plots show the average win rate over supervised fine-tuning (SFT), the Null Rate given an incorrect answer (I.A.) and the Null Rate given a correct answer (C.A.).

## H    Additional Experiments for LLMs: AE-DPO on Offline Datasets

Given the use of offline data, we do not need an oracle during training. We evaluated the AE-DPO approach on a higher number of data points and also against more baselines. To reduce the computational cost of these additional experiments, we applied Qlora to the models used here. In order to evaluate whether our method is able to improve the selection of datapoints in RLHF, we conduct a set of experiments in which we train LLMs on three datasets using one of four methods. The goal of our empirical study is to see whether improving the data selection strategy causes the downstream policy to perform better on a given training task. In order to isolate the effect of the data selection method, we vary the selection method while largely holding our model and training procedure consistent. In all the experiments in this section, we compare four methods: **DPOAE**, the method we presented in Section E; **USDPO**, which chooses $x$ that maximize variance of the log probabilities of completions; **DPO**, the method from Rafailov et al. (2023), selecting batches uniformly at random; and **SFT**, which continues supervised learning with uniformly selected batches. In our training pipeline, we first train a baseline model with a Llama-7B (Touvron et al., 2023) architecture using supervised fine-tuning (SFT) on a 40% split of data. We add a dropout layer before the penultimate linear layer for our uncertainty estimation mechanism and fine tune with dropout active. Next, we train using each of the four methods for 100000 samples, evaluating every 2048 samples—each time using our initial SFT trained model as a starting point. We give additional information on our experimental procedures in Section K.

We evaluate these methods on three different datasets. The first two, the Anthropic Helpful-Harmless (HH) dataset (Bai et al., 2022) and the Stanford Human Preferences (SHP) dataset (Ethayarajh et al., 2022), are taken from the literature. We evaluate policies trained on both

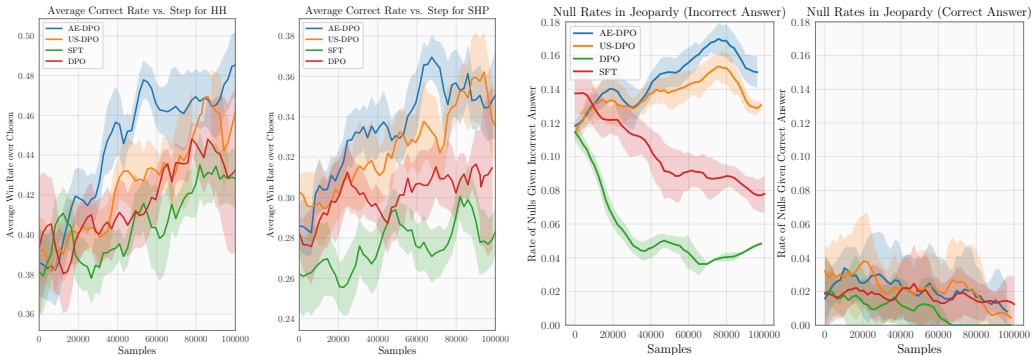

Figure 7: From left: smoothed win rates against preferred choices in dataset of samples generated from each policy at end of RLHF training runs across the final four evaluations, and all seeds, on the HH (first) and SHP (second) datasets. In the latter two plots, we force each policy to generate a (non-null) answer, and then, conditional on the answer being correct (fourth) or incorrect (third), plot the rate at which each policy abstains.

of these by checking the rate at which the policy produces answers which are preferred to the chosen completion for the prompt in the dataset.

For the completions generated from the HH and SHP prompts, we use GPT-3.5 (Brown et al., 2020) to generate winners amongst comparisons between the preferred choices given in the dataset. We give the prompts we use for evaluation in Section J. In Figure 2, we see that for the completions in the later part of our training run, AE-DPO performs best among the methods and outperforms US-DPO as well as the other baselines that sample uniformly. We believe this to be due to our acquisition function $\alpha$, which accounts for the structure of the decision making problem in choosing which point to query. We do find our results to be noisy—due to the computational expense of these trials, we were not able to run each experimental baseline for a large number of seeds to further reduce uncertainty in our results.

For the Jeopardy dataset, Similar to AE-Borda-DPO We found that our models do not learn to provide correct answers at a higher rate through a small amount of DPO training or additional SFT beyond what is required for them to answer the questions. We include an additional exhibit where we use the factual nature of this dataset to begin to evaluate the dropout-based uncertainty estimation techniques we use in Appendix K.3.

For the Jeopardy! dataset, we checked the probability of an empty generation and whether it was the most likely token. We generated a nonempty sequence in order to see whether the generated answer was correct, including as a counterfactual in the cases where the method would have abstained. We plot this in Figure 7, where we see that the AE-DPO method is the only method that learns to abstain from answering questions (modestly) more often when the model would have given the incorrect answer. We also find that the standard DPO method quickly learns not to abstain. No methods abstain more than a couple percent of the time in the case where they would have been correct.

For Jeopardy! we also plot the correctness of the policy over time in Figure 8 which shows that no model substantially learns new factual information. Though this is part of the goal of the agent in the Jeopardy! dataset, note that it is not the entire optimization objective, as we show in Figure 7. Here, it is clear that no policy is able to improve at predicting correct answers on the test set. This is unsurprising as trivia is a difficult generalization problem.

# I The Jeopardy! preference dataset

We generated a set of plausible wrong answers for the Jeopardy! dataset from Huggingface (Wolf et al., 2023) by asking GPT-3.5 for a plausible wrong answer given the question, category, and answer. We found that both the category and correct answer were necessary

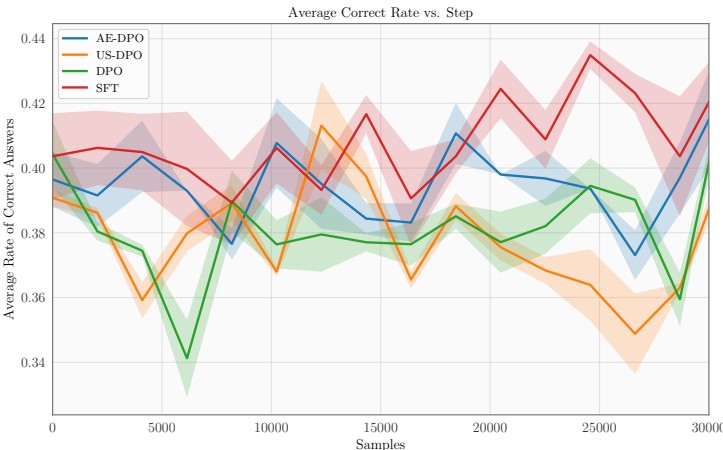

Figure 8: Rate of correct answers for Jeopardy! over time.

```
[System]
You are an assistant coming up with plausible but incorrect answers
 to Jeopardy questions (just the answer, no "what is"). Here's an
example:\n
Q: 'For the last 8 years of his life, Galileo was under house
arrest for espousing this man's theory'
Category: HISTORY
Correct Answer: Copernicus\n
Response: Brahe
[User]
Q: {question}
Category: {category}
Correct Answer: {answer}
Response:
```

Figure 9: The prompt used to collect plausible wrong answers for Jeopardy! questions.

to include to direct GPT-3.5 to generate an answer which was appropriate for the category and to prevent it from accidentally generating a correct answer. We give the prompt used for this process in Figure 9.

## J  Prompt templates

The prompt templates for GPT-4 as the pairwise comparison evaluation judge and GPT-3.5 as the Jeopardy! single answer correctness judge are listed in Figures 10 and 11. We maintain the standardized prompts proved to be effective by Zheng et al. (2023).

## K  Additional Experiment Details

### K.1  Experiment details for AE-Borda-DPO evaluation

We train our initial SFT models for 1 epoch for all datasets. We use the models at their full capacity without any quantization. Further, we use dropout probability of $p = 0.05$, policy constraint weight $\gamma = 0.1$, an uncertainty bonus $\beta = 4$, a learning rate of $5 \times 10^{-7}$, an unlabeled batch size of 96, and a training batch size $b$ of 32. Our implementation was built atop the one provided by the authors of the DPO paper (Rafailov et al., 2023).

```
[System]
Please act as an impartial judge and evaluate the quality of the
responses provided by two AI assistants to the user question
displayed below. You should choose the assistant that follows the
user's instructions and answers the user's question better. Your
evaluation should consider factors such as the helpfulness,
relevance, accuracy, depth, creativity, and level of detail of
their responses. Avoid any position biases and ensure that the
order in which the responses were presented does not influence your
 decision. Do not allow the length of the responses to influence
your evaluation. Do not favor certain names of the assistants. Be
as objective as possible. Output your final verdict by strictly
following this format: 'A' if assistant A is better, 'B' if
assistant B is better, and 'C' for a tie. Output only that
character and do not include any other characters or spaces.

[User Question]
{question}

[The Start of Assistant A's Answer]
{answer_a}
[The End of Assistant A's Answer]

[The Start of Assistant B's Answer]
{answer_b}
[The End of Assistant B's Answer]
```

Figure 10: The default prompt for pairwise comparison.

```
[System]
You are a judge on whether a contestant answer to Jeopardy is
correct given a correct answer. If you don't see the correct answer
 it is not correct. Answer 'Yes' or 'No' is sufficient. Please don'
t use any other words.

[The Start of Correct Answer]
{correct_answer}
[The End of Correct Answer]

[The Start of Contestant Answer]
{contestant_answer}
[The End of Contestant Answer]
```

Figure 11: The default prompt for evaluating single Jeopardy! answer.

### K.2 Experiment details for AE-DPO evaluation

We train our initial SFT models for 1 epoch on the SHP and HH dataset and 2 epochs on the new Jeopardy! dataset. We select the initial training period based on the amount of training after which we obtained a validation loss which had plateaued. We also find it reasonable to add a dropout layer before the penultimate linear layer since we find that adding a dropout layer not to negatively affect the performance in the SFT phase. To aid in fitting the model on our GPUs, we use QLoRa (Hu et al., 2021; Dettmers et al., 2023) with 4bit quantization for model weights and optimize using the 8-bit Lion optimizer (Chen et al., 2023). For the methods with a reference model, we put the policy and the reference model on two separate GPUs. Further, we use dropout probability of $p = 0.05$, policy constraint weight $\gamma = 0.1$, an uncertainty bonus $\beta = 4$, a learning rate of $5 \times 10^{-7}$, an unlabeled batch size of 128, and a training batch size $b$ of 32.

### K.3 Evaluating dropout-based LLM uncertainty estimation

We believe that in general the estimation of uncertainty for LLMs is an important topic of research and progress there will facilitate a more efficient and informed use of this technology. As we discussed in Section 2 and Section 5, we use a dropout-based uncertainty estimation technique to inform the active exploration in this work. Over the course of this study, we considered ensembles and epistemic networks (Osband et al., 2022) as alternative methods for estimating the uncertainty of LLMs. However, each of these methods comes with some additional GPU memory requirement. For epistemic networks, the additional network parameters take GPU memory, while for ensembles, the memory is required to store multiple copies of a network or at least mutiple LoRAs. In our initial studies we found epistemic networks and dropout to perform comparably well and therefore chose dropout due to its smaller memory consumption and good performance. In this section, we explore whether the uncertainties predicted by our estimates differ when the model predicts the correct, incorrect, or null answer and whether these predictions differ in the cases when the model decides to predict null. To do this, we evaluated the log probabilities predicted by $\pi_{\text{SFT}}$ on a test set of 20,560 Jeopardy! clues for the correct, incorrect, and null answer. We computed the sample variances over the log probabilities $\sigma^2(a \mid x) = \sum_{t_i \in a} \sigma^2(t_i \mid x, t_1, \ldots, t_{i-1})$ and plotted their densities in Fig. 12.

We see that the model predicts the highest variances for the log probabilities of incorrect answers. We also see that the the model seems to predict especially low variances for the null token when it decides to output it. The correct answer seems to have a lower variance when the model is willing to predict an answer. We see that the log probabilities of incorrect answers always have a high variance, indicating high uncertainty. We also see that the null token has a low variance when the model has a non-null output indicating certainty that it should not abstain. The variance further drops when it outputs null, indicating certainty about not knowing an answer. The correct answer has a lower variance than the incorrect answer when the model does not abstain. The relative variances of these two curves support that the model provides meaningful indications of uncertainty. Additionally, in the case where the model abstains, even the correct answer has a high variance, indicating a high uncertainty. We believe that these results support that the uncertainty function is at least correlated with the model's knowledge about the input. This offers support to the hypothesis that our estimates of the variance are somewhat meaningful. However, we believe that this is an important research topic and warrants substantial further study under a variety of lenses. We hope that this work will encourage further research in this area.

## L Intransitivity and Condorcet paradox

It is important to note that intransitivity and the presence of Condorcet paradoxes—as discussed in (Liu et al., 2025; Duan et al., 2017) are important considerations when using preference-based methods. Our proposed Borda function is rooted in the classical Borda score/count method and therefore inherently overcomes these issues (Heckel et al., 2019). The Borda score aggregates over pairwise comparisons, which helps mitigate the impact of

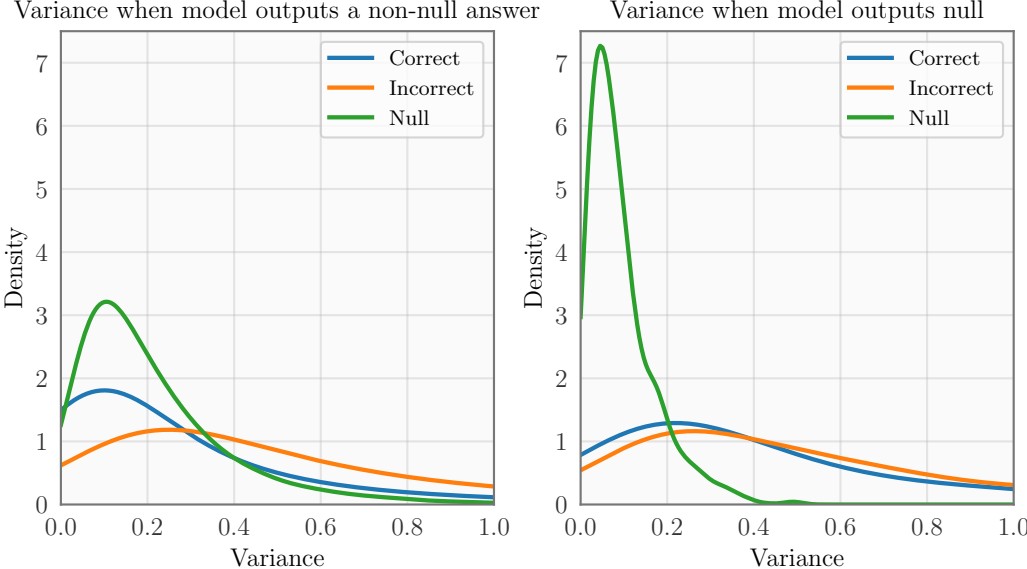

Figure 12: Density of $\sigma(a \mid x)$ conditioned on correct, incorrect, and null values for $a$. The left hand plot depicts the variance distributions conditional on the model outputing a non-null completion, while the right hand is conditional on a null completion.

local inconsistencies or cycles that may arise due to intransitivity. Because the contextual Borda function marginalises over all opponents, it should be immune to cycles in the comparison graph and is always defined, even when pairwise preferences are intransitive (Shah & Wainwright, 2018).

As for the Condorcet paradox, whenever a Condorcet winner exists, it would maximise the Borda function (Van Newenhizen, 1992); otherwise, the maximizer should provide the standard compromise used in duelling-bandit regret. While the Borda rule is not immune to the Condorcet paradox, it is known to produce a robust ranking in practice (Shah & Wainwright, 2018; Heckel et al., 2019). Recent related work points out that while the Condorcet winner may not always exist, the Borda winner is always well-defined (Suk & Agarwal, 2024). We view the use of Borda-based selection as a pragmatic and computationally efficient solution, particularly suitable for the early stages of exploration when uncertainty is high and perfect transitivity is unlikely.

