# OpenReview forum: "Sample Efficient Preference Alignment in LLMs via Active Exploration"
_colmweb.org/COLM/2025/Conference — COLM 2025_

### Official Review · Reviewer_1Thk · 2025-05-11

**Rating:** 8
**Confidence:** 3
**Ethics Flag:** 1

**Summary:**

This paper focuses on efficiently collecting human preference data for training large language models (LLMs). The problem is framed as an active contextual dueling bandit task, and the proposed algorithm is a new exploration algorithm that selectively chooses when and where to request preferences. Theoretical guarantees, including a polynomial worst-case regret bound, back the proposed approach.

The method is further extended to online and offline settings and is applied to preference alignment tasks using several LLMs across four datasets, including two new ones they introduce. The method outperforms baselines with fewer preference samples.

Overall, this work makes a first move in this direction with both a practical and theoretical pursuit to improve data efficiency in preference-based learning for LLMs.

**Questions To Authors:**

1. The propose algorithm leverages Contextual Borda Function for the active exploration procedure, there are always concerns in the efficiency of active exploration due to the existence of Condorcet winner and 'intransitivity' as described below, I wonder does the proposed method has any potential impact or connection to these sophistications?

Condorcet paradox in aligning LLMs: (K. Liu, et al) https://arxiv.org/abs/2503.10990

Intransitivity in preference dataset: (J. Duan, et al) https://arxiv.org/abs/2409.19325

**Reasons To Accept:**

- The related work coverage is comprehensive and navigates the technical contribution, in comparison to Dueling Bandits, Active Contextual Bandit Optimization, alternative contextual bandit methods, uncertainty estimation in large language models, and concurrent work on active data selection in LLMs.
- A wide range of results (3 SFT models) and generalization ability are presented, providing sufficient evidence for the proposed method's success.
- I'm not in a position to verify all the theoretical contributions supporting the arguments. However, the experiment design, presentations, and the story explaining the behavior of the proposed algorithm, e.g., Algorithm 1-2, Figure 2-5, and Figure 12, are easy to understand from a non-technical perspective and are much appreciated, in my opinion.

**Reasons To Reject:**

- I find the arguments, evidences, presentation has high quality from a non-technical perspective.
- I have no concern in the general quality, but in case there are technical problems in the theoretical part of this paper, the accept/rejection decisions should be carefully considered.

---

> ### Author Response · Authors · 2025-05-30
> **Rebuttal**
>
> Thank you for your constructive feedback and for supporting our paper.
>
> ### **Intransitivity and Condorcet paradox**
>
> Thank you for raising this important point. We agree that intransitivity and the presence of Condorcet paradoxes—as discussed in Liu et al. (2024) and Duan et al. (2024)—are important considerations when using preference-based methods.
> Our proposed Borda function is rooted in the classical Borda score/count method and therefore inherently overcomes these issues [3].
> The Borda score aggregates over pairwise comparisons, which helps mitigate the impact of local inconsistencies or cycles that may arise due to intransitivity. Because the contextual Borda function marginalises over all opponents, it should be immune to cycles in the comparison graph and is always defined, even when pairwise preferences are intransitive [2].
>
> As for the Condorcet paradox, whenever a Condorcet winner exists, it would maximise the Borda function [1]; otherwise, the maximizer should provide the standard compromise used in duelling-bandit regret. While the Borda rule is not immune to the Condorcet paradox, it is known to produce a robust ranking in practice [2, 3]. Recent related work points out that while the Condorcet winner may not always exist, the Borda winner is always well-defined [4].
>
> [1] Van Newenhizen, Jill. "The Borda method is most likely to respect the Condorcet principle." Economic Theory 2.1 (1992): 69-83.
>
> [2] Shah, Nihar B., and Martin J. Wainwright. "Simple, robust and optimal ranking from pairwise comparisons." Journal of machine learning research 18.199 (2018): 1-38.
>
> [3] Heckel, Reinhard, et al. "Active ranking from pairwise comparisons and when parametric assumptions do not help." The Annals of Statistics 47(6):3099-3126
>
> [4] Suk, Joe, and Arpit Agarwal. "Non-Stationary Dueling Bandits Under a Weighted Borda Criterion." TMLR (2024).
>
> We view the use of Borda-based selection as a pragmatic and computationally efficient solution, particularly suitable for the early stages of exploration when uncertainty is high and perfect transitivity is unlikely.
>
> We appreciate the references and agree that further analysis of how intransitivity and preference cycles would be a valuable direction for future work. We will include the paragraphs above and the references in our work discussion.

---

> > ### Comment · Reviewer_1Thk · 2025-06-08
> >
> > Thank you for clarifying the position of the proposed technique and for your candid response to my questions. I maintain my score.

---

> > > ### Author Response · Authors · 2025-06-10
> > > **Additional Results**
> > >
> > > Thank you for your constructive feedback. As suggested during the review process, we conducted additional experiments using the OAT baseline on four of our tasks, using the Pythia-2.8B model across all four datasets. The results can be found at the following link: https://anonymous.4open.science/r/AE_BORDA_DPO-D1DC/rebuttal_exp.pdf .We collected as many results as time permitted and are currently running the OAT baseline on the remaining tasks. These additional results will be included in the camera-ready version of the paper. In all completed experiments, our approach consistently outperforms the OAT baseline. We sincerely appreciate the valuable suggestions from the review process and believe that these additions meaningfully strengthen our submission. If our rebuttal and the new results address your concerns, we would be grateful if you would consider raising your score.

---

### Official Review · Reviewer_up2F · 2025-05-12

**Rating:** 5
**Confidence:** 3
**Ethics Flag:** 1

**Summary:**

The data efficiency in alignment has been a long-term problem. To release the burden of human annotation, authors propose a novel active data selection process based on active contextual dueling bandit. The core of the algorithm is the contextual Borda function, which is estimated by kernelized ridge regression. Both context and one action are selected based on the Borda function. The former one is selected by uncertainty, and the latter one is selected by finding the argmax of upper confidence bounds. The authors also provided theoretical bounds as well as practical implementation for DPO. Results show that AE-Borda performs well on uniform sampling.

**Reasons To Accept:**

* The research question is important and interesting
* The method is overall sound
* The method is novel
* The authors release a new dataset which may be useful for further research.

**Reasons To Reject:**

* The paper is a bit hard to understand as many terms are not explained very clearly, requiring readers to read extra materials outside of the paper. For example, the paper will be much clearer if the authors can explain RKHS and Li et al. (2023) mentioned in Line 153 more.

* The experiment does not compare other active learning methods (like some mentioned in the Appendix). Therefore, it is unclear how well the method is in general sense.

* During evaluation, authors compare two methods against a fixed answer, such as the SFT answer, and do not compare the two methods directly, which is more direct.

---

> ### Author Response · Authors · 2025-05-30
> **Rebuttal**
>
> Thank you for your constructive feedback. Below we address your concerns.
>
> ### **Clarity of the paper**
>
> In line 153, we aimed to include a succinct description of the sampling rule in Li et al. (2023). We will elaborate further on the details and include the following paragraph in the final version of the paper:
>
> "We build upon the sampling strategy introduced in Li et al. (2023) and adapt it to our setting. Their algorithm uses uncertainty estimates over the Q-function to guide exploration. Specifically, at each decision step, the algorithm selects the state where the uncertainty in the value of the best action is highest. Formally, the state maximizes the gap between optimistic and pessimistic Q-value estimates. Then, it chooses the action with the highest optimistic Q-value at that state. This approach focuses exploration on parts of the state space where the agent is most uncertain about which action is best, promoting efficient learning of a near-optimal policy."
>
> For the RKHS, we have included a detailed definition of RKHS regression in Appendix A, and a detailed analysis of the impact of different RKHS norms in Appendix C.
>
> ### **On baseline active learning methods**
>
> In this work, we proposed two algorithms. For the first, AE-BORDA-DPO, we employed an oracle during both training and evaluation. Due to the cumulative cost of the oracle queries, we limited our comparison to a single baseline. The second algorithm, AE-DPO (described in the appendix), was evaluated on synthetic datasets using the oracle only during evaluation. Because this incurred a lower cost, we were able to include a comparison with an uncertainty-based baseline.
>
> We agree that additional comparisons for AE-BORDA-DPO would strengthen the paper. We are currently implementing the algorithm proposed in Liu et al. (suggested by reviewer JaHM) in our codebase to enable a fair comparison. We will share the results as soon as they are available and include them in the rebuttal if ready in time.
>
> ### **Comparisons against a fixed generation**
>
> Thank you for the suggestion. To the best of our knowledge, in prior work on DPO and alignment, the standard evaluation approach is to compare all methods against a fixed generation. Since all algorithms are evaluated against the same SFT generation, this allows for fair and consistent evaluation, yielding insights equivalent to direct pairwise comparisons. Another reason we chose this strategy was that we believed direct pairwise comparisons across multiple algorithms (as we add baselines) would introduce complexity and impact the readability of the results.

---

> > ### Comment · Reviewer_up2F · 2025-06-10
> >
> > Sorry for the late reply. I read the rebuttal and thank the authors for the rebuttal.
> >
> > I believe it is still important to compare directly for fine-grained granularity. Since the extra active learning results are not available yet, I decided to maintain my current score.

---

> > > ### Author Response · Authors · 2025-06-10
> > > **Additional Results**
> > >
> > > Thank you for your constructive feedback. As suggested during the review process, we conducted additional experiments using the OAT baseline on four of our tasks, using the Pythia-2.8B model across all four datasets. The results can be found at the following link: https://anonymous.4open.science/r/AE_BORDA_DPO-D1DC/rebuttal_exp.pdf .We collected as many results as time permitted and are currently running the OAT baseline on the remaining tasks. These additional results will be included in the camera-ready version of the paper. In all completed experiments, our approach consistently outperforms the OAT baseline. We sincerely appreciate the valuable suggestions from the review process and believe that these additions meaningfully strengthen our submission. If our rebuttal and the new results address your concerns, we would be grateful if you would consider raising your score.

---

### Official Review · Reviewer_JaHM · 2025-05-13

**Rating:** 6
**Confidence:** 3
**Ethics Flag:** 1

**Summary:**

This paper consider the active contextual dueling bandit problem in the context of LLM alignment, where the system can control both the prompts and the model responses presented to a human expert. The authors propose a UCB-style algorithm aimed at achieving sample-efficient alignment, by adaptively selecting prompts and responses to obtain informative human labels. Experimental results shows improved performance over baseline methods under the same query budget.

**Questions To Authors:**

Dropout-based uncertainty estimation is known to be unreliable in large-scale deep models (Lakshminarayanan et al. 2017). Have you evaluated the quality of uncertainty estimates? This seems critical to assess whether the observed sample-efficiency improvements are indeed due to uncertainty-guided exploration.

Lakshminarayanan, Balaji, Alexander Pritzel, and Charles Blundell. "Simple and scalable predictive uncertainty estimation using deep ensembles." Advances in neural information processing systems 30 (2017).

**Reasons To Accept:**

The topic of this paper is interesting and important. The paper addresses a practically setting for LLM preference alignment, where the human expert provides feedback and the prompts are controllable. This contrasts with the more commonly studied online serving scenario, where prompts are user-generated and not under the model designer's control. Designing algorithms tailored to different operational constraints is essential, particularly in alignment tasks, where human feedback is expensive and the scale of deployment can be large.

The paper is generally well-written and easy to follow. The figures are informative and help clarify the method.

**Reasons To Reject:**

The paper omits comparison to some relevant prior work. For instance, Liu et al. consider a similar setting and propose methods to improve sample efficiency under both exploration–exploitation and pure exploration setting. The pure exploration setting in Liu et al. seems to share the same objective as this paper. Moreover, the proposed algorithm in this work shares a similar motivation and structure. Since this paper assumes control over prompts, which is a stronger assumption than in Liu et al., I would expect a direct comparison, and possibly better performance.

Liu, Zichen, et al. "Sample-efficient alignment for llms." arXiv preprint arXiv:2411.01493 (2024).

---

> ### Author Response · Authors · 2025-05-30
> **Rebuttal**
>
> Thank you for your positive and constructive feedback on our paper, and for pointing us to the relevant references.
>
> ### **Comparison with Liu, Zichen, et al.**
>
> In this work, we proposed two algorithms. For the first, AE-BORDA-DPO, we employed an oracle during both training and evaluation. Due to the cumulative cost of the oracle queries, we limited our comparison to a single baseline. The second algorithm, AE-DPO (described in the appendix), was evaluated on synthetic datasets using the oracle only during evaluation. Because this incurred a lower cost, we were able to include a comparison with an uncertainty-based baseline.
>
> We agree that additional comparisons for AE-BORDA-DPO would strengthen the paper. We are currently implementing the algorithm proposed in Liu et al. in our codebase to enable a fair comparison. We will share the results as soon as they are available and will aim to include them in the rebuttal if ready in time.
>
> ### **Uncertainty estimation for LLMs**
>
> We agree that the estimation of uncertainty for LLMs is an important topic. Over the course of this study, in addition to dropout, we considered ensembles and epistemic networks (Osband et al., 2022) as alternative methods for estimating the uncertainty. However, each of these methods comes with some additional GPU memory requirements. In our initial studies, we found epistemic networks and dropout to perform comparably well and therefore chose dropout due to its smaller memory consumption and good performance. In section K.3 of the appendix, we provide a study and comparison of the two types of uncertainty to justify the use of dropout in this context.  We also provide a discussion in the related work, section D of the appendix. Importantly, in our setting, uncertainty estimates are used primarily to guide exploration. The consistent performance improvements observed across multiple experiments suggest that even approximate uncertainty estimates can be effective for sample selection in practice.

---

> > ### Author Response · Authors · 2025-06-10
> > **Additional Results**
> >
> > Thank you for your constructive feedback. As suggested during the review process, we conducted additional experiments using the OAT baseline on four of our tasks, using the Pythia-2.8B model across all four datasets. The results can be found at the following link: https://anonymous.4open.science/r/AE_BORDA_DPO-D1DC/rebuttal_exp.pdf .We collected as many results as time permitted and are currently running the OAT baseline on the remaining tasks. These additional results will be included in the camera-ready version of the paper. In all completed experiments, our approach consistently outperforms the OAT baseline. We sincerely appreciate the valuable suggestions from the review process and believe that these additions meaningfully strengthen our submission. If our rebuttal and the new results address your concerns, we would be grateful if you would consider raising your score.

---

### Decision · Program_Chairs · 2025-07-08

**Decision:**

Accept

**Comment:**

This paper proposes a theoretically grounded and practically motivated active learning method for sample-efficient preference alignment in LLMs, framed via a contextual dueling bandit formulation. It introduces novel Borda-based selection strategies and demonstrates consistent improvements over baselines on multiple datasets. The paper is generally well-written, and the authors have responded thoughtfully to reviewer concerns, including additional experiments. While comparisons to some recent related work were initially lacking, the rebuttal and added results sufficiently address this. Given the solid theoretical contributions, practical significance, and overall positive reviewer consensus (scores 6, 5, 8), I recommend Accept.